# Population-based trends in hospitalizations due to injection drug use-related serious bacterial infections, Oregon, 2008 to 2018

Jeffrey Capizzi[1], Judith Leahy[1], Haven Wheelock[2], Jonathan Garcia [3], Luke Strnad[4], Monica Sikka[4], Honora Englander[4], Ann Thomas [1], P. Todd Korthuis[1,4°], Timothy William Menza [1,4°]*

**1** Public Health Division, Oregon Health Authority, Portland, Oregon, United States of America, **2** OutsideIn, A Federally Qualified Health Center (FQHC), Portland, Oregon, United States of America, **3** College of Public Health and Human Sciences, Oregon State University, Corvallis, Oregon, United States of America, **4** Department of Medicine, Oregon Health & Science University, Portland, Oregon, United States of America

☺ These authors contributed equally to this work.
* TIMOTHY.W.MENZA@dhsoha.state.or.us

## Abstract

### Background

Injection drug use has far-reaching social, economic, and health consequences. Serious bacterial infections, including skin/soft tissue infections, osteomyelitis, bacteremia, and endocarditis, are particularly morbid and mortal consequences of injection drug use.

### Methods

We conducted a population-based retrospective cohort analysis of hospitalizations among patients with a diagnosis code for substance use *and* a serious bacterial infection during the same hospital admission using Oregon Hospital Discharge Data. We examined trends in hospitalizations and costs of hospitalizations attributable to injection drug use-related serious bacterial infections from January 1, 2008 through December 31, 2018.

### Results

From 2008 to 2018, Oregon hospital discharge data included 4,084,743 hospitalizations among 2,090,359 patients. During the study period, hospitalizations for injection drug use-related serious bacterial infection increased from 980 to 6,265 per year, or from 0.26% to 1.68% of all hospitalizations (*P*<0.001). The number of unique patients with an injection drug use-related serious bacterial infection increased from 839 to 5,055, or from 2.52% to 8.46% of all patients (*P*<0.001). While hospitalizations for all injection drug use-related serious bacterial infections increased over the study period, bacteremia/sepsis hospitalizations rose most rapidly with an 18-fold increase. Opioid use diagnoses accounted for the largest percentage of hospitalizations for injection drug use-related serious bacterial infections, but hospitalizations for amphetamine-type stimulant-related serious bacterial infections rose most rapidly with a 15-fold increase. People living with HIV and HCV experienced increases

**Data Availability Statement:** The data underlying the results presented in the study are available from the Office of Health Analytics of the Oregon

Health Authority, https://www.oregon.gov/oha/
HPA/ANALYTICS/Pages/Hospital-Reporting.aspx.

**Funding:** This work was supported by grants from
the NIH National Institute on Drug Abuse
(UH3DA044831, U01TR002631, UG1DA015815)
to PTK. The URL for the National Institute on Drug
Abuse is drugabuse.gov. The funder had no role in
study design, implementation, analysis, or
manuscript review.

**Competing interests:** The authors have declared
that no competing interests exist.

in hospitalizations for injection drug use-related serious bacterial infection during the study
period. Overall, the total cost of hospitalizations for injection drug use-related serious bacte-
rial infections increased from $16,305,129 in 2008 to $150,879,237 in 2018 ($P$<0.001).

## Conclusions

In Oregon, hospitalizations for injection drug use-related serious bacterial infections increased
dramatically and exacted a substantial cost on the health care system from 2008 to 2018. This
increase in hospitalizations represents an opportunity to initiate substance use disorder treat-
ment and harm reduction services to improve outcomes for people who inject drugs.

## Introduction

The substance use disorder (SUD) epidemic has long been a major public health challenge in
Oregon, which led the nation in prescription opioid prescribing in 2006, and experienced
increases in opioid overdose early in the epidemic [1]. Oregon experienced a substantial
increase in the rate of opioid use diagnoses among the state's Medicaid population from 101
cases per 100,000 population in 2005 to 506 cases per 100,000 population in 2015 (Personal
communication, John W. McIlveen, Ph.D., State Opioid Treatment Authority, Health Systems
Division, Oregon Health Authority, September 12, 2017). While deaths due to prescription
opioids have declined in the setting of reductions in provider prescribing, non-pharmaceutical
fentanyl deaths have increased from 0.4 per 100,000 population in 2009 to 1.8 per 100,000 pop-
ulation in 2018 [2]. Concurrently, and consistent with national data, methamphetamine use,
alone and in combination with opioids, has increased significantly statewide [3–5]. Metham-
phetamine-related deaths increased from 0.5 per 100,000 population in 2006–2008 to 3.4 per
100,000 population in 2015–2017 [2].

The substance use disorder (SUD) epidemic has led to increased infectious complications
of injection drug use (IDU) [6], including serious bacterial infections (SBIs), such as skin and
soft tissue infections (SSTI), bacteremia or sepsis, osteomyelitis, infectious endocarditis, as
well as hepatitis B virus (HBV), hepatitis C virus (HCV), and HIV infections [7–10]. IDU-
related SBIs are associated with high morbidity and mortality [11] and may be an important
marker of SUD severity; those with an IDU-related SBI experienced a more than fifty-fold
increase in overdose death compared to those without an IDU-related SBI [12]. Hospitaliza-
tion rates and hospitalization costs associated with IDU-related SBIs are important measures
of the social, economic, and public health burden of IDU. These data highlight the critical
need for SUD screening, harm reduction services, and patient engagement–all interventions
that can and should happen at both the hospital- and community-level [13].

Despite recent reports of increased hospitalizations for SBIs [8, 10, 14, 15], few studies pro-
vide population-based estimates of hospitalization trends and costs among people who inject
drugs (PWID). A recent statewide assessment of endocarditis hospitalizations in North Caro-
lina documented a more than twelve-fold increase in hospitalizations for heart valve replace-
ment among patients with diagnoses of drug use and endocarditis between 2007 and 2017
[10]. Such estimates of hospitalizations for IDU-related SBI and their costs are urgently needed
to optimize resource allocation to clinical and public health interventions best suited to limit-
ing the infectious disease consequences of IDU.

In addition, increases in infectious endocarditis and acute HCV diagnoses may indicate
new or emerging IDU networks, especially in areas where IDU has not been a previously estab-
lished problem [14, 16]. These emerging networks may experience greater vulnerability to
HIV and HCV transmission [16]. In Oregon, the proportion of new HIV infections

attributable to IDU increased from 11% in 2012 to 26% in 2018 [17]. Concurrent increases in chronic HCV infection among those younger than 30 years of age reflect more recent infection likely acquired in the context of IDU [18]. Thus, SBI associated with IDU may indicate concurrent or future increases in HIV and HCV infection.

In the context of the trends in Oregon SUD metrics and the increasing infectious complications related to IDU in other jurisdictions and their implications for HIV and HCV transmission, we sought to 1) describe statewide trends in IDU-related SBI hospitalizations overall and by SBI type and drug use diagnosis, 2) assess IDU-related SBI diagnoses among individuals living with HIV and HCV, and 3) and determine the annual costs of IDU-related SBI overall and by SBI type.

## Methods

### Study design and population

We conducted a population-based retrospective cohort study of people hospitalized in Oregon with IDU-related SBIs from January 1, 2008 to December 31, 2018. These hospitalizations included residents of other states receiving care in Oregon.

### Data source and setting

We used the Oregon Hospital Discharge Dataset (HDD) containing all inpatient hospitalizations from 60 Oregon hospitals, excluding only the state's two Veteran's Affairs hospitals and one long-term acute care facility [19]. Thirty-three hospitals (55%) were located in rural or frontier areas. We accessed the Oregon HDD through a data use agreement with the Office of Health Analytics of the Oregon Health Authority (OHA). Data were de-identified after matching to Oregon HIV and HCV surveillance data (defined below) but before any further analysis. Representatives of the OHA Science and Epidemiology Council deemed this work public health practice and exempt from institutional review board review based on Oregon Administrative Rule 333-019-0005 [20].

### Case classification

We defined an IDU-related SBI hospitalization as any hospital stay in which a patient received a diagnosis of substance use associated with opioids, cocaine, amphetamine-type stimulants (ATS), sedatives or other drugs and at least one of the following infections: endocarditis, bacteremia/sepsis, osteomyelitis, or SSTI. We vetted and modified the algorithm based on *International Classification of Diseases*, 9th and 10th Revisions (ICD-9 and ICD-10), codes through a key informant process that included consultation from the Centers for Disease Control and Prevention, wound clinic and emergency department doctors, and a medical information specialist. The algorithm excluded diagnostic codes not specific for drug use or infections associated with IDU, such as codes reflecting drug-use remission or poisoning. We expected this algorithm to be highly specific and less sensitive relative to the actual number of hospitalizations among PWID because of incomplete coding of drug use [21]. Between January 1, 2008 and September 30, 2015 we used ICD-9 diagnostic codes to classify cases. Subsequent to September 30, 2015 HDD implemented ICD-10 codes which we used for case classification. S1 Data includes diagnosis codes used for inclusion in and, exclusion from, the case classification.

To capture the full extent of the impact of IDU-related SBI, we did not restrict our analysis by age. During the study period, there were 34 patients less than twelve years of age included in the dataset; 22 were neonates and twelve were one to eleven years of age. Together, they represented 0.1% (34/34,404) unique patients in the dataset. While SBI hospitalizations in these

age groups are not necessarily the result of IDU directly, they may be the indirect result of caregiver IDU [22].

To identify patients living with either HIV or HCV, we matched patients in the HDD to Oregon HIV and HCV surveillance data using both deterministic (SPSS, IBM, Armonk, NY) and probabilistic (LinkPlus, Centers for Disease Control and Prevention, Atlanta, GA) methods based on first, middle, last name, alias names, and date of birth. Ninety percent (4,093/ 4,527) of persons in the HDD diagnosed with HIV matched to reported HIV cases, and 75% (17,127/22,795) of persons diagnosed with HCV matched to reported HCV cases. Patients in the HDD with HIV who were not matched to the HIV surveillance system were individuals who never resided in Oregon and were, thus, never reported as confirmed cases in Oregon. Matching of HDD patients with HCV to the surveillance system was less successful since HCV surveillance in Oregon is collected passively by laboratory reporting and does not document aliases.

## Cost determination

We determined cost in 2018 dollars by adjusting the charged amount by the cost-to-charge ratios published by the OHA Office of Health Analytics [23]. There was a different adjustment for each hospital and year. For each year, we calculated the total costs and the median and interquartile range (IQR) of costs per IDU-related SBI hospitalization overall and for each infection type: endocarditis, bacteremia/sepsis, osteomyelitis, or SSTI.

## Statistical analysis

We summarized demographic characteristics of gender, age, race/ethnicity, HIV and HCV status, Oregon residence, and geography (urban, rural/frontier defined by zip code [24]) overall, by SBI type, and by drug class used. We calculated the annual number of hospitalizations for SBIs for each year. For hospitalizations coded with more than one SBI, we prioritized the condition of greatest severity in order to prevent duplicating cost or counts across SBIs [21]. We prioritized endocarditis, then osteomyelitis (both potential complications of bacteremia) followed by bacteremia/sepsis, and SSTI. For example, we consider a patient diagnosed with endocarditis and a SSTI during the same admission as a single hospitalization. As the care delivered during such a hospitalization is likely driven by the endocarditis diagnosis, we attribute the cost of the hospitalization to endocarditis rather than the SSTI (based on the severity hierarchy) and not to both diagnoses (which would result in cost duplication). For hospitalizations coded with more than one drug class, we created an additional category to distinguish polysubstance use from single-class drug use. We then calculated the total IDU-related SBI hospitalization costs and median and IQR of costs per IDU-related SBI hospitalization by year and SBI type. For comparison, we calculated the total costs and the median and IQR of costs per SBI hospitalization per year among patients without a concurrent drug use diagnosis code and who never had a drug use diagnosis code during the follow-up period.

We used generalized linear models with a log link, a Poisson distribution and robust standard errors to perform tests-of-trend in IDU-related SBI hospitalizations and costs over time. We defined statistical significance at the $P<0.05$ level and used SPSS for all analyses (IBM, Armonk, NY).

## Results

Oregon discharge data for the period of January 1, 2008 through December 31, 2018 included 4,084,743 hospitalizations among 2,090,359 unique patients. Ninety-three percent of hospitalizations reflected care to Oregon residents.

## Demographics of patients hospitalized for injection drug use-related serious bacterial infections

Our algorithm identified 34,404 individuals hospitalized at least once for an IDU-related SBI between January 1, 2008 and December 31, 2018. Fifty-four percent were men, 49% were over the age of 50, 84% were white, 3% identified as Hispanic, 72% lived in urban areas, and 95% were Oregon residents (Table 1). Over half had a diagnosis code of opioid-only use, one-quarter had a diagnosis code of ATS-only use, and 16% had diagnosis codes of polysubstance use, most commonly combination opioid and ATS use. Seven percent were living with HIV and 38% were living with HCV.

Patients hospitalized for SSTI and osteomyelitis were more likely to be men than women (Table 2). Thirty-one percent of those hospitalized for bacteremia or sepsis and thirty-eight percent of those hospitalized for endocarditis were aged 60 years or older. Forty-seven percent of those with SSTI and 41% of those with osteomyelitis were living with HCV infection. The percentage of patients with a diagnosis of opioid-only use was greatest among those with osteomyelitis, while the percentage of patients diagnosed with ATS-only use was highest among those hospitalized for SSTI.

**Table 1. Sociodemographic characteristics of patients hospitalized with an injection drug use-related serious bacterial infection, Hospital Discharge Data, Oregon, 2008–20018.**

| | | N = 34,404, n (%) |
|---|---|---|
| **Sex** | Female | 15,865 (46) |
| **Age, years** | 0–11 | 34 (<1) |
| | 12–19 | 207 (1) |
| | 20–29 | 4,329 (13) |
| | 30–39 | 6,271 (18) |
| | 40–49 | 6,618 (19) |
| | 50–59 | 7,906 (23) |
| | 60–69 | 5274 (15) |
| | 70–79 | 2358 (7) |
| | 80 and older | 1407 (4) |
| **Race/ethnicity** | Hispanic/Latinx | 1,112 (3) |
| | Black | 1,186 (3) |
| | White | 28,869 (84) |
| | Other[a] | 1,993 (6) |
| | Refused/Unknown | 1,244 (4) |
| **HIV status** | HIV case | 2,360 (7) |
| **HCV status** | HCV case | 12,902 (38) |
| **Drug use diagnosis** | Opioid-only | 17,807 (52) |
| | Amphetamine-type stimulants-only | 8,497 (25) |
| | Cocaine-only | 548 (2) |
| | Sedative-only | 304 (1) |
| | Other drug | 1,827 (5) |
| | More than one class | 5,421 (16) |
| **Oregon residence** | Oregon resident | 32,720 (95) |
| **Geography** | Rural/Frontier | 8,098 (25) |
| | Urban | 24,622 (75) |

[a] Other includes American Indian/Alaska Native, Asian, and Native Hawaiian/Pacific Islander

**Table 2. Sociodemographic characteristics by type of injection drug-use related serious bacterial infection, Hospital Discharge Data, Oregon, 2008–2018.**

| | | SSTI (N = 11,515) | Bacteremia (N = 16,166) | Osteomyelitis (N = 2,196) | Endocarditis (N = 4,527) |
|---|---|---|---|---|---|
| | | n (%) | n (%) | n (%) | n (%) |
| **Sex** | Female | 4,701 (41) | 8,010 (50) | 787 (36) | 2,367 (52) |
| | Male | 6,814 (59) | 8,156 (50) | 1,409 (64) | 2,160 (48) |
| **Age, years** | 0–11 | 4 (<1) | 28 (<1) | 1 (<1) | 1 (<1) |
| | 12–19 | 87 (1) | 99 (1) | 6 (<1) | 15 (<1) |
| | 20–29 | 1,694 (15) | 1,918 (12) | 144 (7) | 573 (13) |
| | 30–39 | 2,639 (23) | 2,543 (16) | 347 (16) | 742 (16) |
| | 40–49 | 2,761 (24) | 2,753 (17) | 481 (22) | 623 (14) |
| | 50–59 | 2,643 (23) | 3,755 (23) | 676 (31) | 832 (18) |
| | 60–69 | 1163 (10) | 2963 (18) | 381 (17) | 767 (17) |
| | 70–79 | 355 (3) | 1376 (9) | 126 (6) | 501 (11) |
| | 80 and older | 169 (1) | 731 (5) | 34 (2) | 473 (10) |
| **Race/ethnicity** | Hispanic/Latinx | 305 (3) | 580 (4) | 89 (4) | 138 (3) |
| | Black | 339 (3) | 562 (3) | 59 (3) | 226 (5) |
| | White | 9,477 (82) | 13,743 (85) | 1,871 (85) | 3,778 (83) |
| | Other[a] | 862 (7) | 806 (5) | 104 (5) | 221 (5) |
| | Refused/Unknown | 532 (5) | 475 (3) | 73 (3) | 164 (4) |
| **HIV case** | Not HIV case | 10,727 (93) | 15,138 (94) | 2,024 (92) | 4,155 (92) |
| | HIV case | 788 (7) | 1,028 (6) | 172 (8) | 372 (8) |
| **HCV case** | Not HCV case | 6,132 (53) | 11,037 (68) | 1,295 (59) | 3,038 (67) |
| | HCV case | 5,383 (47) | 5,129 (32) | 901 (41) | 1,489 (33) |
| **Drug use** | Opioid-only | 5,700 (50) | 8,640 (53) | 1,211 (55) | 2,256 (50) |
| | Amphetamine-type stimulants-only | 2,968 (26) | 3,968 (25) | 487 (22) | 1,074 (24) |
| | Cocaine-only | 167 (1) | 239 (1) | 32 (1) | 110 (2) |
| | Sedative-only | 40 (<1) | 192 (1) | 11 (<1) | 61 (1) |
| | Other drug | 727 (6) | 761 (5) | 143 (7) | 196 (4) |
| | More than one class | 1,913 (17) | 2,366 (15) | 312 (14) | 830 (18) |
| **Geography** | Rural/Frontier | 2,301 (21) | 4,009 (26) | 568 (27) | 1,220 (28) |
| | Urban | 8,626 (79) | 11,404 (74) | 1,528 (73) | 3,064 (72) |

[a] Other includes American Indian/Alaska Native, Asian, and Native Hawaiian/Pacific Islander

Sixty-four percent and 65% of those hospitalized for an IDU-related SBI with a diagnosis of ATS-only use and cocaine-only use, respectively, were men (Table 3). Thirty-nine percent and 62% of patients hospitalized for an IDU-related SBI in the context of a diagnosis of opioid-only use and sedative-only use, respectively, were aged 60 years or older. Thirty-nine percent of those hospitalized for an IDU-related SBI with a diagnosis of cocaine-only use were Black. Eleven percent and 59% of patients hospitalized with an IDU-related SBI and a diagnosis of polysubstance use were living with HIV and living with HCV, respectively. Thirty-three percent of those with an IDU-related SBI hospitalization and an ATS-only use diagnosis were living in rural/frontier areas.

## Overall time trends in injection drug use-related serious bacterial infections

IDU-related SBI hospitalizations increased over six-fold, from 980 to 6,265 hospitalizations, or 0.26% to 1.68% of all hospitalizations from 2008 to 2018 (*P*<0.001; Fig 1). The number of

**Table 3. Sociodemographic characteristics of patients hospitalized for injection drug use-related serious bacterial infection by substance use diagnosis, Hospital Discharge Data, Oregon, 2008–2018.**

| | | Opioid-only (N = 17,807) | Amphetamine-type stimulants-only (N = 8,497) | Cocaine-only (N = 548) | Sedative-only (N = 304) | Other drug (N = 1,827) | More than one class (N = 5,421) |
|---|---|---|---|---|---|---|---|
| | | n (%) | n (%) | n (%) | n (%) | n (%) | n (%) |
| **Sex** | Female | 9,128 (51) | 3,039 (36) | 193 (35) | 198 (65) | 827 (45) | 2,480 (46) |
| | Male | 8,679 (49) | 5,458 (64) | 355 (65) | 106 (35) | 1,000 (55) | 2,941 (54) |
| **Age, years** | 0–11 | 29 (<1) | 0 | 0 | 2 (<1) | 1 (<1) | 2 (<1) |
| | 12–19 | 44 (<1) | 84 (1) | 6 (1) | 1 (<1) | 19 (1) | 53 (1) |
| | 20–29 | 1,640 (9) | 993 (1) | 65 (12) | 15 (5) | 241 (13) | 1,375 (25) |
| | 30–39 | 2,514 (14) | 1,748 (21) | 76 (14) | 11 (4) | 347 (19) | 1,575 (29) |
| | 40–49 | 2,641 (15) | 2,328 (27) | 120 (22) | 30 (10) | 414 (23) | 1,085 (20) |
| | 50–59 | 3,989 (22) | 2,351 (28) | 167 (30) | 57 (19) | 424 (23) | 918 (17) |
| | 60–69 | 3664 (21) | 886 (10) | 96 (18) | 73 (24) | 220 (12) | 335 (6) |
| | 70–79 | 2047 (11) | 99 (1) | 15 (3) | 45 (15) | 95 (5) | 57 (1) |
| | 80 and older | 1239 (7) | 8 (<1) | 3 (1) | 70 (23) | 66 (4) | 21 (<1) |
| **Race/ ethnicity** | Hispanic/ Latinx | 491 (3) | 324 (4) | 22 (4) | 10 (3) | 73 (4) | 192 (4) |
| | Black | 473 (3) | 215 (3) | 216 (39) | 2 (1) | 61 (3) | 219 (4) |
| | White | 15,135 (85) | 7,254 (85) | 215 (39) | 276 (91) | 1,492 (82) | 4,497 (83) |
| | Other[a] | 1,037 (6) | 460 (5) | 63 (11) | 9 (3) | 115 (6) | 309 (6) |
| | Refused/ Unknown | 671 (4) | 244 (3) | 32 (6) | 7 (2) | 86 (5) | 204 (4) |
| **HIV status** | Not HIV case | 16,850 (95) | 7,803 (92) | 511 (93) | 300 (99) | 1,729 (95) | 4,851 (89) |
| | HIV case | 957 (5) | 694 (8) | 37 (7) | 4 (1) | 98 (5) | 570 (11) |
| **HCV status** | Not HCV case | 11,838 (66) | 5,558 (65) | 416 (76) | 280 (92) | 1,200 (66) | 2,210 (41) |
| | HCV case | 5,969 (34) | 2,939 (25) | 132 (24) | 24 (8) | 627 (34) | 3,211 (59) |
| **Geography** | Rural/ Frontier | 3,838 (23) | 2,662 (33) | 39 (7) | 86 (30) | 532 (31) | 941 (18) |
| | Urban | 13,175 (77) | 5,391 (67) | 486 (93) | 204 (70) | 1,206 (69) | 4,160 (82) |

[a] Other includes American Indian/Alaska Native, Asian, and Native Hawaiian/Pacific Islander

unique persons with an IDU-related SBI increased from 839 to 5,055, or from 2.52% to 8.46% of all unique persons hospitalized during this period ($P<0.001$).

## Trends in injection drug use-related serious bacterial infection by infection type and drug use

IDU-related bacteremia/sepsis hospitalizations increased 18-fold, from 189 to 3,345, or from 0.05% to 0.9% of all hospitalizations from 2008 to 2018 ($P<0.001$; Fig 1). Endocarditis hospitalizations increased eight-fold, from 112 to 929, or from 0.03% to 0.5% of all hospitalizations ($P<0.001$). Osteomyelitis hospitalizations increased six-fold, from 59 to 371, or from 0.02% to 0.1% of all hospitalizations ($P<0.001$) while SSTI hospitalizations increased three-fold, from 620 to 1620, or from 0.16% to 0.43% of all hospitalizations ($P<0.001$).

From 2008 to 2018, opioids were the most common drug class associated with hospitalization for IDU-related SBI (52%), followed by ATS (25%), multiple drugs (16%), cocaine (2%), and sedatives (1%). The largest increase in IDU-related SBI between 2008 and 2018 occurred among people diagnosed with ATS-only use (a 15-fold increase, $P<0.001$). The next largest

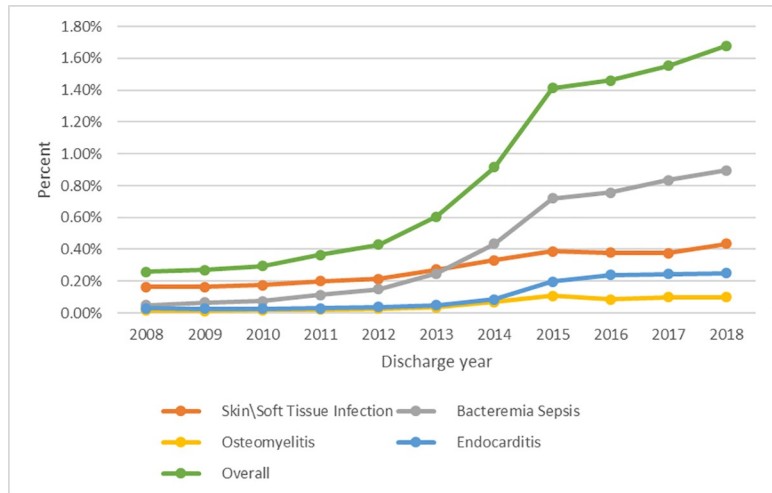

**Fig 1. Injection drug use-related SBI hospitalizations, overall and by SBI type, as a percentage of all hospitalizations, Hospital Discharge Data, Oregon, 2008–2018.**

increases occurred among those with polysubstance use (13-fold increase, $P<0.001$) and using sedative-only use diagnoses (12-fold increase, $P<0.001$). In comparison, IDU-related SBI hospitalizations among those with an opioid-only use diagnosis showed a five-fold increase ($P<0.001$, Fig 2).

## Trends in injection drug use-related serious bacterial infections among those living with HIV and HCV

IDU-related SBI hospitalizations among persons with HIV increased five-fold, from 18 to 174, or an increase from 1.7% to 13.0% of all hospitalizations among people living with HIV ($P<0.001$; Fig 3). IDU-related SBI hospitalizations among persons with HCV also increased five-fold, from 105 to 910, or an increase from 3.7% to 17.0% of all hospitalizations among people living with HCV ($P<0.001$).

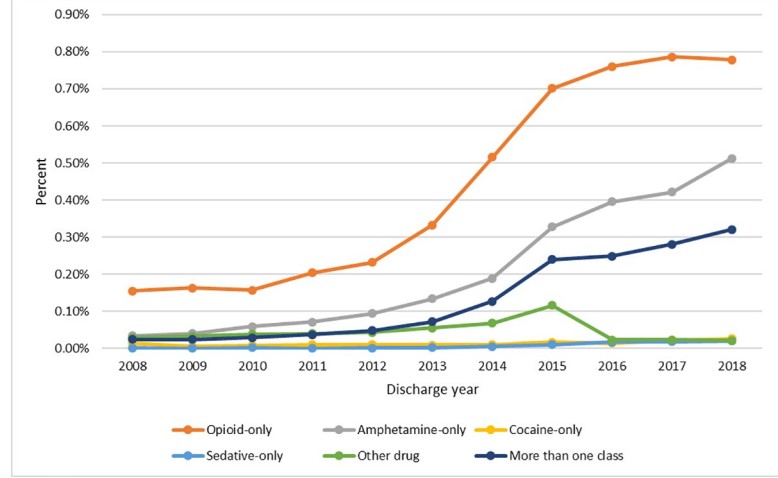

**Fig 2. Injection drug use-related SBI hospitalizations, by drug type, as a percentage of all hospitalizations, Hospital Discharge Data, Oregon, 2008–2018.**

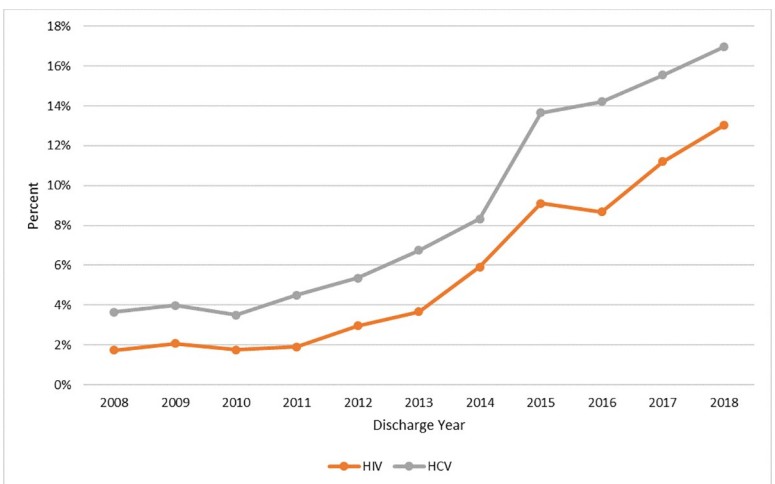

**Fig 3. Injection drug use-related SBI hospitalizations among people living with HIV and HCV as a percentage of all hospitalizations among people living with HIV and HCV, respectively, Hospital Discharge Data, Oregon, 2008–2018.**

## Cost associated with injection drug use-related serious bacterial infections

Overall, total costs of IDU-related SBI hospitalizations increased nine-fold from $16,305,129 in 2008 to $150,879,237 in 2018 ($P<0.001$; Table 4). The median cost per IDU-related SBI hospitalization increased from $9,525 (IQR: $5,814, $18,441) in 2008 to $13,000 (IQR: $7,809, $24,671) in 2018. Increases in total cost were greatest among bacteremia/sepsis hospitalizations (a 15-fold increase from $5,622,041 to $83,845,107, $P<0.001$) followed by endocarditis (a 14-fold increased from $2,229,222 to $30,366,521, $P<0.001$), osteomyelitis (an eight-fold increase from $1,701,881 to $13,015,821, $P<0.001$), and SSTI (a 3.5-fold increase from $6,751,985 to $23,651,787, $P<0.001$, Tables 5 and 6).

Total costs of SBI hospitalizations that were not IDU-related were greater than IDU-related SBI hospitalizations due to the greater number of SBI hospitalizations that were not IDU-related (Tables 5 and 6). In 2018, there were 53,497 SBI hospitalizations that were not IDU-related

**Table 4. Total costs of injection drug use-related serious bacterial infections and median cost per hospitalization for serious bacterial infections, by injection drug use, Hospital Discharge Data, Oregon, 2008–2018[a].**

| Year | SBI, IDU-related | | | | SBI, not IDU-related | | | |
|---|---|---|---|---|---|---|---|---|
| | Total | Median | 25%ile | 75%ile | Total | Median | 25%ile | 75%ile |
| 2008 | $16,305,129 | $9,525 | $5,814 | $18,441 | $697,939,223 | $11,210 | $6,352 | $23,121 |
| 2009 | $17,689,550 | $9,857 | $6,005 | $18,537 | $773,443,861 | $11,950 | $6,732 | $24,905 |
| 2010 | $18,426,648 | $9,982 | $6,117 | $17,280 | $821,897,961 | $12,305 | $6,916 | $25,348 |
| 2011 | $20,888,954 | $10,123 | $6,295 | $16,962 | $855,136,477 | $12,471 | $7,043 | $25,460 |
| 2012 | $23,130,549 | $9,885 | $6,128 | $16,089 | $878,329,637 | $12,366 | $7,021 | $24,926 |
| 2013 | $33,692,406 | $10,409 | $6,496 | $17,348 | $928,546,762 | $12,642 | $7,313 | $25,121 |
| 2014 | $58,821,754 | $10,725 | $6,840 | $18,777 | $1,042,552,439 | $12,487 | $7,298 | $24,819 |
| 2015 | $121,564,949 | $12,991 | $7,818 | $24,784 | $1,159,360,623 | $12,735 | $7,527 | $25,262 |
| 2016 | $142,651,621 | $13,629 | $8,315 | $27,083 | $1,192,774,951 | $13,510 | $7,876 | $26,560 |
| 2017 | $147,094,529 | $13,937 | $8,356 | $26,978 | $1,251,412,186 | $13,663 | $8,013 | $27,167 |
| 2018 | $150,879,237 | $13,000 | $7,809 | $24,671 | $1,192,496,100 | $12,914 | $7,675 | $24,704 |

[a]Costs adjusted by charge-to-cost ratio (by year and hospital) and adjusted to 2018 US dollars

Table 5. Total costs of IDU-related skin/soft tissue infection and bacteremia and median cost per hospitalization for skin/soft tissue infection and bacteremia, by injection drug use, Hospital Discharge Data, Oregon, 2008–2018[a].

| Year | SSTI, IDU-related | | | | SSTI, not IDU-related | | | | Bacteremia, IDU-related | | | | Bacteremia, not IDU-related | | | |
|---|---|---|---|---|---|---|---|---|---|---|---|---|---|---|---|---|
| | Total | Median | 25%ile | 75%ile | Total | Median | 25%ile | 75%ile | Total | Median | 25%ile | 75%ile | Total | Median | 25%ile | 75%ile |
| 2008 | $6,751,985 | $7,804 | $5,116 | $12,235 | $116,174,137 | $7,933 | $4,873 | $14,353 | $5,622,041 | $19,868 | $10,782 | $33,968 | $343,451,334 | $15,319 | $8,181 | $33,928 |
| 2009 | $6,953,456 | $8,238 | $5,278 | $12,455 | $122,461,812 | $8,355 | $5,099 | $15,041 | $6,974,869 | $16,476 | $9,433 | $33,775 | $398,393,335 | $15,808 | $8,422 | $34,001 |
| 2010 | $7,439,488 | $8,183 | $5,295 | $13,214 | $123,166,582 | $8,488 | $5,172 | $14,767 | $6,636,342 | $14,418 | $7,746 | $23,376 | $440,862,732 | $15,828 | $8,759 | $32,729 |
| 2011 | $7,758,188 | $7,894 | $5,317 | $12,166 | $117,533,145 | $8,487 | $5,225 | $15,444 | $8,777,363 | $13,407 | $8,465 | $22,718 | $460,225,911 | $15,114 | $8,433 | $30,916 |
| 2012 | $8,629,763 | $8,263 | $5,474 | $13,135 | $115,605,435 | $8,723 | $5,385 | $15,439 | $8,802,159 | $11,724 | $7,149 | $18,197 | $489,908,831 | $13,799 | $7,967 | $27,639 |
| 2013 | $10,596,151 | $8,483 | $5,499 | $12,838 | $115,500,784 | $8,932 | $5,594 | $15,710 | $15,843,281 | $12,396 | $7,519 | $19,187 | $534,438,487 | $13,662 | $8,058 | $26,601 |
| 2014 | $14,479,263 | $8,513 | $5,678 | $12,913 | $117,626,589 | $8,675 | $5,567 | $15,245 | $29,790,498 | $11,703 | $7,561 | $20,402 | $607,854,320 | $13,213 | $7,834 | $25,508 |
| 2015 | $19,310,106 | $9,220 | $5,950 | $14,751 | $116,446,173 | $8,776 | $5,596 | $15,465 | $65,503,907 | $14,216 | $8,505 | $26,540 | $609,513,162 | $12,859 | $7,805 | $24,773 |
| 2016 | $19,676,208 | $9,493 | $6,011 | $15,131 | $118,161,184 | $9,414 | $5,830 | $16,772 | $80,432,942 | $15,174 | $9,330 | $29,083 | $639,658,241 | $13,548 | $8,139 | $25,793 |
| 2017 | $21,202,658 | $9,811 | $6,386 | $16,040 | $123,953,721 | $9,588 | $6,018 | $17,167 | $81,144,765 | $14,857 | $9,046 | $28,678 | $680,094,936 | $13,504 | $8,169 | $25,624 |
| 2018 | $23,651,787 | $9,194 | $5,962 | $15,836 | $121,319,280 | $9,414 | $5,841 | $16,745 | $83,845,107 | $13,798 | $8,415 | $24,891 | $660,489,907 | $12,703 | $7,781 | $23,418 |

[a]Costs adjusted by charge-to-cost ratio (by year and hospital) and adjusted to 2018 US dollars

Table 6. Sum of total costs of IDU-related osteomyelitis and endocarditis and median cost per hospitalization for osteomyelitis and endocarditis, by injection drug use, Hospital Discharge Data, Oregon, 2008–2018[a].

| Year | Osteomyelitis, IDU-related | | | | Osteomyelitis, not IDU-related | | | | Endocarditis, IDU-related | | | | Endocarditis, not IDU-related | | | |
|---|---|---|---|---|---|---|---|---|---|---|---|---|---|---|---|---|
| | Total | Median | 25%ile | 75%ile | Total | Median | 25%ile | 75%ile | Total | Median | 25%ile | 75%ile | Total | Median | 25%ile | 75%ile |
| 2008 | $1,701,881 | $19,302 | $11,199 | $32,220 | $39,242,787 | $15,886 | $9,240 | $28,099 | $2,229,222 | $11,329 | $6,362 | $23,497 | $199,070,966 | $10,756 | $6,309 | $22,092 |
| 2009 | $1,287,714 | $16,257 | $9,631 | $29,579 | $43,937,442 | $15,993 | $9,743 | $28,531 | $2,473,510 | $13,205 | $6,112 | $33,352 | $208,651,271 | $11,513 | $6,559 | $24,366 |
| 2010 | $2,027,157 | $19,470 | $11,044 | $39,966 | $45,757,096 | $17,754 | $10,337 | $30,456 | $2,323,662 | $13,329 | $6,922 | $26,031 | $212,111,551 | $11,727 | $6,569 | $25,485 |
| 2011 | $2,186,453 | $21,036 | $11,845 | $37,818 | $53,913,613 | $17,645 | $10,368 | $31,965 | $2,166,949 | $16,261 | $8,140 | $24,798 | $223,463,808 | $12,020 | $6,850 | $25,945 |
| 2012 | $2,568,475 | $16,109 | $9,534 | $33,693 | $52,795,599 | $17,792 | $10,652 | $30,607 | $3,130,153 | $13,133 | $7,086 | $22,269 | $220,019,772 | $12,728 | $6,865 | $29,635 |
| 2013 | $3,274,987 | $18,830 | $11,163 | $34,254 | $57,048,009 | $17,645 | $10,901 | $29,703 | $3,977,986 | $13,977 | $7,560 | $24,237 | $221,559,482 | $13,264 | $7,124 | $32,668 |
| 2014 | $6,326,910 | $18,435 | $10,444 | $29,011 | $58,908,996 | $17,559 | $10,784 | $29,485 | $8,225,082 | $13,551 | $8,231 | $27,513 | $258,162,535 | $13,820 | $7,495 | $32,600 |
| 2015 | $13,542,693 | $20,673 | $12,689 | $39,435 | $61,451,050 | $18,381 | $11,273 | $31,035 | $23,208,243 | $16,576 | $8,977 | $38,005 | $371,950,238 | $14,979 | $8,253 | $33,473 |
| 2016 | $11,436,419 | $24,558 | $12,066 | $44,108 | $43,668,130 | $18,885 | $11,926 | $33,056 | $31,106,052 | $17,839 | $9,400 | $39,494 | $391,287,396 | $15,789 | $8,691 | $35,289 |
| 2017 | $13,249,918 | $22,182 | $13,824 | $45,977 | $53,574,358 | $19,646 | $11,520 | $33,457 | $31,497,189 | $16,972 | $9,832 | $38,536 | $393,789,172 | $16,687 | $8,955 | $38,416 |
| 2018 | $13,015,821 | $21,609 | $12,193 | $42,091 | $54,978,748 | $17,708 | $11,278 | $30,646 | $30,366,521 | $17,557 | $9,638 | $38,462 | $355,708,164 | $15,575 | $8,521 | $35,085 |

[a]Costs adjusted by charge-to-cost ratio (by year and hospital) and adjusted to 2018 US dollars

compared to 6,265 IDU-related SBI hospitalizations. However, there was only a 1.7-fold increase in total costs of SBI hospitalizations that were not IDU-related over the study period; the largest increase was an almost 2-fold increase in total costs of non-IDU-related bacteremia hospitalizations. In 2018, the median cost per IDU-related SBI hospitalization was similar to the median cost of non-IDU-related SBI hospitalizations. Also, the median cost per hospitalization for IDU-related SSTI was similar to the median cost per hospitalization for SSTI that was not IDU-related. In contrast, the median cost per hospitalization for IDU-related bacteremia, osteo-myelitis, and endocarditis was greater than the median cost per hospitalization for bacteremia, osteomyelitis, and endocarditis that was not IDU-related. Osteomyelitis accounted for the great-est difference in median cost per IDU-related ($21,609) versus non-IDU-related ($17,708) SBI hospitalization.

## Discussion

Our study demonstrates increasing hospitalizations and costs for IDU-related SBI, underscor-ing the substantial impact of the SUD epidemic on healthcare and public health systems.

IDU-related bacteremia/sepsis increased dramatically during the study period, ultimately comprising the largest proportion of hospitalizations and incurring the largest total costs in subsequent years. Consistent with data from North Carolina, we also observed a significant increase in endocarditis diagnoses [10]. However, the magnitude of IDU-related cases was greater in Oregon and, in 2018, the 929 cases of IDU-related endocarditis cost the healthcare system over $30 million dollars. Increases in endocarditis may signal emerging IDU networks in need of harm reduction services [14]. Our data suggest, however, that bacteremia may be a more sensitive indicator of emerging IDU networks given its greater incidence compared to endocarditis. Use of bacteremia as such an indicator may also be more useful in rural areas with lower population density. In contrast, osteomyelitis and endocarditis may be markers of more severe, long-standing SUD characterized by more frequent use and riskier injection practices [12, 25]. The increasing proportion of hospitalizations for IDU-related SBI among people living with HIV and HCV likely reflects an increase in IDU-related risk behavior that may indicate ongoing community-level transmission of HCV and HIV.

While opioid-only use diagnoses accounted for the greatest proportion of IDU-related SBI overall during the study period, we observed the largest increase in IDU-related SBI among people with ATS-only use diagnoses. This increase in SBI mirrors state-level data indicating an increasing rate of both fatal and non-fatal overdoses related to methamphetamine use during the same time period [2], increased ATS-related hospitalizations and costs [3], and increased treatment admissions for concomitant methamphetamine and opioid use disorder in Oregon and nationally [4, 5]. Among PWID participating in the Centers for Disease Control and Pre-vention's National Behavioral Surveillance (NHBS) survey in Portland, OR, combination methamphetamine and opioid use was associated with more frequent injection, syringe shar-ing, and decreased use of syringe exchange, factors which may increase the risk of SBI (per-sonal communication, Timothy Menza, Principal Investigator, Portland NHBS, July 2, 2020). In addition to SBI, methamphetamine use is associated with recent increases in HIV infection, early syphilis, and HCV among PWID [18, 26, 27]. Thus, inpatient admissions for SBI also represent key opportunities for HIV, STI, and HCV screening, prevention and linkage to care [28]. Our findings also emphasize a critical need for the incorporation of addiction medicine training into the fields of infectious diseases and hospital medicine [29, 30].

Our study has several limitations. First, the study relies on administrative claims data which may not accurately capture diagnoses and other demographics (e.g. race, ethnicity). The algo-rithm we used to identify IDU-related SBIs required a patient to have both a drug use and

serious bacterial infection diagnosis coded during the same stay. This conservative definition may underestimate the true burden and costs of IDU-related SBI as patients had multiple visits with a drug use code <u>or</u> infection before having both during the same hospitalization. This pattern implies that the algorithm was more specific and less sensitive, and that many patients with only an SBI diagnosis may have a drug use disorder that was not coded during the hospitalization. We suspect that the algorithm also underestimated the proportion of SBI associated with polysubstance use. Among PWID participating in NHBS in Portland, OR, 70% of respondents reported using both ATS and opioids (personal communication, Timothy Menza, Principal Investigator, NHBS Portland, October 7, 2020). In contrast, only 16% of SBI hospitalizations in the current study were associated with more than one drug use code. Second, the study period bridged the transition from the use of ICD-9 to ICD-10 diagnosis codes. Trends among diagnoses were consistent across this transition but demonstrated more variation as providers started using ICD-10. Third, these data reflect the socio-demographic composition, patterns of drug use and hospital care in Oregon all of which may differ from other regions of the United States. Our study sample, like the population of Oregon, was predominantly white (75.1% of the population of Oregon identifies as white), which makes detecting differences in IDU-related SBI hospitalizations by race and ethnicity challenging. However, we observed that 5% of endocarditis hospitalizations occurred among Black patients, but Black individuals comprise only 2% of the Oregon population. Thus, compared to their representation in the general population, Black patients are likely over-represented among those diagnosed with IDU-related endocarditis in Oregon. The observed age distribution of our sample is also a potential source of bias. A quarter of our sample was over the age of 60. Older patients may use prescription medications that fall within the categories of the drug codes of interest and may be more likely to experience SBIs [31, 32]. Thus, older patients may be more likely to be misclassified as having an IDU-related SBI compared to younger patients. Furthermore, hospitalizations among older patients may be more costly than hospitalizations experienced by younger patients [33].

Despite these limitations, our study has important implications for health systems, payers, and policy-makers. Increases in hospitalizations underscores the opportunity to initiate substance use disorder treatment and harm reduction services during hospitalization. While many people may not seek addictions care in the community, hospitalization can present a reachable moment to engage non-treatment seeking adults in care [34]. Hospital-based initiation of buprenorphine and methadone are acceptable to patients and providers [13, 35]. Hospital-based SUD treatment can reduce post-hospital substance use [36], increase engagement in post-hospital SUD treatment [13], and improve care quality [37, 38]. Evidence is mixed as to effects of hospital-based addictions care on readmissions for SBI [13, 34, 39–41].

From 2008 to 2018, we observed greater increases in total costs associated with hospitalizations for IDU-related SBI, overall and for each SBI type, compared to SBI hospitalizations that were not IDU-related. In 2018, the median cost per hospitalization for bacteremia, osteomyelitis, and endocarditis that were IDU-related was greater than the median cost per hospitalization for these infections that were not IDU-related. The reason for the greater cost of hospitalization for these infection types among PWID may be three-fold. First, providers may not consider PWID candidates for outpatient parenteral antibiotic therapy (OPAT), thus, prolonging hospital lengths of stay for infections that may require 2–6 weeks of intravenous antimicrobials. Although emerging data indicate that injection drug use is not a contraindication to OPAT and that PWID can be successfully and safely treated in the outpatient setting, these data had likely not yet impacted practice in our study period [42, 43]. Second, due to fear of discrimination and judgment by healthcare providers, PWID with SBI may present to care later than people who do not inject drugs [44]. The resulting delay may result in more

complicated presentations requiring higher levels of care and surgical intervention for source control. Finally, PWID may be more likely to experience infections with methicillin-resistant *Staphylococcus aureus* (MRSA) [45] which often has more severe clinical presentations that require the use of more costly therapeutics [46].

The cost of increased hospitalizations represents a growing preventable burden that should be of major concern to healthcare payers. Keeshin and colleagues noted that an admission for infectious endocarditis could cost more than the start-up costs of a syringe exchange program [14] and Schranz and colleagues estimated that an admission for endocarditis exceeds the cost of a year's worth of medication for opioid use disorder [10]; our data are consistent with these findings. Presentations of our data to local public health authorities, health systems, federally qualified health centers, law enforcement, community members with lived and professional experience with PWID, and other stakeholders have led to the adoption of syringe service programs in several high-needs Oregon counties. Other public health entities looking to expand access to sterile syringes and injection supplies may consider leveraging their own data on IDU-related SBI to promote community-based harm reduction for PWID. Payers might consider funding individual- and community-level preventive strategies and response measures, such as health and human service provider trainings on IDU, sterile injection patient education, increased access to sterile equipment, syringes and naloxone, adult HAV/HBV vaccinations, biomedical HIV prevention, and substance abuse treatment programs could be established or strengthened to decrease IDU-associated infections. Our study shows that payers already have skin in the game; they might consider upstream, preventive efforts to improve health and reduce costs.

While opioid use accounted for over half of IDU-related SBIs during the study period, the greatest increases in hospitalizations and costs for IDU-related SBI were associated with ATS and polysubstance use. The rising rates of hospitalizations and costs related to ATS and polysubstance use suggest that the singular "opioid epidemic" narrative which has driven the public health response to the SUD crisis may miss the mark [4, 5, 18, 47]. Instead, healthcare delivery systems, policy-makers, and researchers must broaden their scope and take into account the importance of polysubstance use across communities.

## Conclusion

IDU-related SBI hospitalizations and costs increased in Oregon between 2008 and 2018. Most IDU-related SBI hospitalizations were associated with opioid-only use diagnoses, but we observed substantial increases in IDU-related SBI associated with ATS-only and polysubstance use diagnoses. Additionally, IDU-related SBI hospitalizations increased among people living with HIV and HCV. Our findings suggest an urgent need to expand community-based addiction treatment and harm reduction services to prevent SBI, and to expand hospital-based addictions care to engage people with SUD during hospitalization.

## Supporting information

**S1 Data. ICD-9 and ICD-10 code equivalents for Hospital Discharge Data: Injection drug use-related serious bacterial infection hospitalizations.**
(DOCX)

## Acknowledgments

We thank the following people for their contributions to creating the algorithm to define injection drug use-related serious bacterial infections: Dr. Tanya Page (Providence Medical Group,

Oregon), Dr. Bill Walter (Lane County Health Department, Oregon), Dr. Svetla Slavova (Associate Professor, University of Kentucky), Theresa Garvin (Director, St Claire Medical Center, Kentucky), and Dr. Jon Zibell (Senior Public Health Analyst at Research Triangle International [RTI] and previously with the Division of Viral Hepatitis, CDC).

## Author Contributions

**Conceptualization:** Judith Leahy, Haven Wheelock, Jonathan Garcia, Luke Strnad, Ann Thomas, P. Todd Korthuis, Timothy William Menza.

**Data curation:** Jeffrey Capizzi.

**Formal analysis:** Jeffrey Capizzi.

**Funding acquisition:** P. Todd Korthuis.

**Investigation:** Ann Thomas.

**Methodology:** Jeffrey Capizzi, Haven Wheelock, Jonathan Garcia, Luke Strnad, Ann Thomas, P. Todd Korthuis, Timothy William Menza.

**Project administration:** Judith Leahy.

**Supervision:** Ann Thomas, P. Todd Korthuis, Timothy William Menza.

**Writing – original draft:** Jeffrey Capizzi, Judith Leahy, Haven Wheelock, Jonathan Garcia, Ann Thomas, P. Todd Korthuis.

**Writing – review & editing:** Jeffrey Capizzi, Monica Sikka, Honora Englander, Ann Thomas, P. Todd Korthuis, Timothy William Menza.

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
