## [Decision Letter · Decision Letter 0]

10 Sep 2020

PONE-D-20-23516

Population-based trends in hospitalizations due to injection drug use-related serious bacterial infections, Oregon, 2008 to 2018

PLOS ONE

Dear Dr. Menza,

Thank you for submitting your manuscript to PLOS ONE. After careful consideration, we feel that it has merit but does not fully meet PLOS ONE’s publication criteria as it currently stands. Therefore, we invite you to submit a revised version of the manuscript that addresses the points raised during the review process.

We look forward to receiving your revised manuscript.

Kind regards,

Nickolas D. Zaller

Academic Editor

PLOS ONE

Journal Requirements:

"This work was supported by grants from the NIH National Institute on Drug Abuse

(UH3DA044831, U01TR002631, UG1DA015815) to PTK. The URL for the National

Institute on Drug Abuse is drugabuse.gov. The funder had no role in study design,

implementation, analysis, or manuscript review."

We note that one or more of the authors are employed by a commercial company: "Outside In"

Reviewers' comments:

Reviewer's Responses to Questions

**Comments to the Author**

1. Is the manuscript technically sound, and do the data support the conclusions?

Reviewer #1: Yes

Reviewer #2: Partly

2. Has the statistical analysis been performed appropriately and rigorously? 

Reviewer #1: Yes

Reviewer #2: Yes

3. Have the authors made all data underlying the findings in their manuscript fully available?

Reviewer #1: No

Reviewer #2: No

4. Is the manuscript presented in an intelligible fashion and written in standard English?

Reviewer #1: Yes

Reviewer #2: Yes

5. Review Comments to the Author

Reviewer #1: The manuscript is clear and succinct in laying out the extent to which serious bacterial infections have increased and placed an increasing burden on hospitals in Oregon. The changes over the 11-year study period are a compelling reminder of the severity of the illicit drug use problem in the US and especially compelling since the population of Oregon is so overwhelmingly White. The changes in drug use patterns is reflected somewhat in the increasing number of infections among individuals whose drug use include amphetamine-type stimulants, and the authors are wise to point this out.

There are some elements of the manuscript that need greater attention from the authors. One important area is the results on costs. Given the increase in cases, the total costs have skyrocketed, but some of the increase appears to be related to the cost for each episode over the 11-year period. It would be instructive if the authors could compare the increasing cost per episode over time to cost per episode for similar infections in which the discharge codes did not include evidence substance use and for the cost per episode for hospitalization costs in general. This context would benefit those in state agencies, insurance companies, and hospital systems have a better understanding of the role of inflationary hospital costs and simultaneously the implications of a failure to prevent increases in serious bacterial infections among people who inject drugs.

The authors can dispense with Figure 1, since it is the cumulative total of the data presented in Figure 2, simply by adding the data on the annual total to Figure 2.

I take issue with the statement in the Conclusion that “SBI hospitalizations and costs increase[s]…were associated with amphetamine and polysubstance use diagnoses and increased hospitalizations among people living with HIV and HCV.” The majority of cases continued to involve opioids, so not mentioning opioids in the conclusion is an unfortunate omission.

There are a number of minor edits that would improve the text. These are referred to in the list below by line number and suggested edits are capitalized in many instances.

Line 50, Abstract: This would be clearer if the text read: “During the study period, hospitalizations…increased from 980 to 6,265 PER YEAR, or from 0.26% to 1.68%...”

Line 76, Introduction: “Methamphetamine-related deaths increasED from 0.5 per 100,000…”

Lines 81-82, Introduction reads: “IDU-related SBIs are associated with high morbidity and mortality with a more than fifty-fold increase in death in some studies.” Increased relative to what?

Line 84-85, Introduction: The sentence that begins, “They also highlight critical opportunities for SUD screening, harm reduction services, and patient engagement…” does not read clearly because the antecedents for “They” is “Hospitalization rates and hospitalization costs”, which aren’t really opportunities. I suggest the sentence be rephrased to read, “They also highlight THE critical NEED for SUD screening, harm reduction services, and patient engagement…”

Line 88, Introduction: This is the first time that the acronym PWID is used and so it should be written out.

Line 116-117, Methods: “We accessed these data…” The previous sentence describes the percentage of hospitals in rural/frontier areas. It is possible to construe that the “these” in the next sentence refers only to these hospitals and not all the other hospitals. Please rephrase for clarity.

Line 125, Methods: I suggest using the term amphetamine-type stimulants rather than amphetamines to describe this class of drugs. The authors might consider then using the acronym ATS for subsequent appearances. This is particularly pertinent since the most common illicit ATS would be methamphetamine,

Line 136, Methods: No need for the commas.

Line 187, Results: Should read “OPIOID and amphetamine use”, not heroin use. Nowhere else in the methods or results are opioids broken down into different compounds, nor do the ICD-9 or ICD-10 codes provide such specificity.

Line 340-341, Discussion: The text refers to “the “opioid epidemic” narrative which has driven the public health response”. It is unclear to me what narrative they are referring to, so the authors need to be much more specific.

Reviewer #2: The submitted article offers interesting insight into the consequences of SBIs among PWID. A particular strength is the use statewide data. With that, my comments are as follows:

Minor comments:

1. Please ensure you define each acronym when it is first used.

2. With the exception of stratifying by age category, you have quite a large sample. Chi-square is very sensitive to large sample sizes. Because of this, the p-values may be meaningless.

3. The cost analysis appears to be very general. While interesting, a deeper look into excess cost attributable to IDU would be far more impactful.

Major comments:

1. In the Statistical Analysis section of the Methods, the explanation for prioritizing SBIs of greatest severity to prevent duplicating cost or counts is not clear. Hospitalizations with multiple SBI would be obscured intentionally then?

2. For your age category variable, why set the cut off at 60? This age group represents a sizable proportion of your sample (26%). It would be good to know what the breakdown is within this category.

3. The last sentence of the second paragraph in the Case Classification section ("While SBI hospitalizations.."), are there any published articles to support this?

4. You indicate that you did not restrict your analysis by age to "capture the full extent of the impact of IDU-related SBI." Further, you offer some deeper justification for including young individuals. However, you do not mention the other end of the age spectrum. We know that older individuals also can be impacted by SBIs unrelated to IDU, and many may have non-illicit, medicinal dependencies. It is possible, if not probable, that a considerable number of older individuals in your sample are misclassified as PWID, yet there is not a single mention of this potential bias. Compounding the issue is that older individuals comprise a large proportion of your sample (26% 60 and older), yet we can't tell if these are mostly people in their 60s, 70s, 80s, etc. All of these issues culminate in a potentially biased cost analysis that likely includes a considerable number of misclassified, older individuals who may incur greater costs in the first place due to infirmity and/or longer lengths of stay.

6. PLOS authors have the option to publish the peer review history of their article (what does this mean?). If published, this will include your full peer review and any attached files.

Reviewer #1: No

Reviewer #2: **Yes: **Michael Cima

---

## [Author Response · Author response to Decision Letter 0]

10 Oct 2020

Journal Requirements:

We have now formatted the manuscript to PLOS ONE’s style requirements.

"This work was supported by grants from the NIH National Institute on Drug Abuse

(UH3DA044831, U01TR002631, UG1DA015815) to PTK. The URL for the National

Institute on Drug Abuse is drugabuse.gov. The funder had no role in study design,

implementation, analysis, or manuscript review."

We note that one or more of the authors are employed by a commercial company: "Outside In"

OutsideIn is a federally-qualified health center (FQHC) and non-profit organization, not a commercial company. OutsideIn is publicly funded and provides safety net services to patients without insurance with a focus on youth and people affected by substance use and houselessness. HW, the author affiliated with Outside In, is the director of OutsideIn’s drug user health program. We now indicate in the author affiliations that OutsideIn is a federally-qualified health center. We do not feel that this affiliation represents a funding source nor a competing interest in the way that a consulting, pharmaceutical or biotechnology company does. Knowing this information, if PLOS ONE still requires amendments to the Funding Statement and Competing Interests Statement, we would be happy to do so. 

Thank you. We now include the captions for supporting information files at the end of the manuscript. 

 

Comments to the Author

Please use the space provided to explain your answers to the questions above. You may also include additional comments for the author, including concerns about dual publication, research ethics, or publication ethics. (Please upload your 

review as an attachment if it exceeds 20,000 characters)

Reviewer #1: The manuscript is clear and succinct in laying out the extent to which serious bacterial infections have increased and placed an increasing burden on hospitals in Oregon. The changes over the 11-year study period are a compelling reminder of the severity of the illicit drug use problem in the US and especially compelling since the population of Oregon is so overwhelmingly White. The changes in drug use patterns is reflected somewhat in the increasing number of infections among individuals whose drug use include amphetamine-type stimulants, and the authors are wise to point this out.

There are some elements of the manuscript that need greater attention from the authors. One important area is the results on costs. Given the increase in cases, the total costs have skyrocketed, but some of the increase appears to be related to the cost for each episode over the 11-year period. It would be instructive if the authors could compare the increasing cost per episode over time to cost per episode for similar infections in which the discharge codes did not include evidence substance use and for the cost per episode for hospitalization costs in general. This context would benefit those in state agencies, insurance companies, and hospital systems have a better understanding of the role of inflationary hospital costs and simultaneously the implications of a failure to prevent increases in serious bacterial infections among people who inject drugs.

We now present the cost data for those who did not have diagnosis codes indicating substance use. We have updated the methods, results, and discussion to reflect the inclusion of these data. 

The authors can dispense with Figure 1, since it is the cumulative total of the data presented in Figure 2, simply by adding the data on the annual total to Figure 2.

We’ve now combined Figures 1 and 2.

I take issue with the statement in the Conclusion that “SBI hospitalizations and costs increase[s]…were associated with amphetamine and polysubstance use diagnoses and increased hospitalizations among people living with HIV and HCV.” The majority of cases continued to involve opioids, so not mentioning opioids in the conclusion is an unfortunate omission.

We’ve addressed this statement in the Conclusion.

There are a number of minor edits that would improve the text. These are referred to in the list below by line number and suggested edits are capitalized in many instances.

Line 50, Abstract: This would be clearer if the text read: “During the study period, hospitalizations…increased from 980 to 6,265 PER YEAR, or from 0.26% to 1.68%...”

Edit incorporated.

Line 76, Introduction: “Methamphetamine-related deaths increasED from 0.5 per 100,000…”

Edit incorporated.

Lines 81-82, Introduction reads: “IDU-related SBIs are associated with high morbidity and mortality with a more than fifty-fold increase in death in some studies.” Increased relative to what?

Thanks for catching this. The clause now reads: “…those with an IDU-related SBI experienced a more than fifty-fold increase in overdose death compared to those without an IDU-related SBI [12].” We have also revised the text to state that IDU-related SBI may be a marker of severe SUD as a reason for the association with overdose death.

Line 84-85, Introduction: The sentence that begins, “They also highlight critical opportunities for SUD screening, harm reduction services, and patient engagement…” does not read clearly because the antecedents for “They” is “Hospitalization rates and hospitalization costs”, which aren’t really opportunities. I suggest the sentence be rephrased to read, “They also highlight THE critical NEED for SUD screening, harm reduction services, and patient engagement…”

The sentence now reads: “These data highlight the critical need for SUD screening, harm reduction services, and patient engagement – all interventions that can and should happen at both the hospital- and community-level [13].”

Line 88, Introduction: This is the first time that the acronym PWID is used and so it should be written out.

Edit incorporated.

Line 116-117, Methods: “We accessed these data…” The previous sentence describes the percentage of hospitals in rural/frontier areas. It is possible to construe that the “these” in the next sentence refers only to these hospitals and not all the other hospitals. Please rephrase for clarity.

We’ve replaced “these data” with “Oregon HDD” to clarify that we are accessing all of the hospital discharge data and not just the rural data.

Line 125, Methods: I suggest using the term amphetamine-type stimulants rather than amphetamines to describe this class of drugs. The authors might consider then using the acronym ATS for subsequent appearances. This is particularly pertinent since the most common illicit ATS would be methamphetamine,

Edit incorporated.

Line 136, Methods: No need for the commas.

Commas deleted.

Line 187, Results: Should read “OPIOID and amphetamine use”, not heroin use. Nowhere else in the methods or results are opioids broken down into different compounds, nor do the ICD-9 or ICD-10 codes provide such specificity.

Good catch. We’ve now edited it to read opioid rather than heroin. 

Line 340-341, Discussion: The text refers to “the “opioid epidemic” narrative which has driven the public health response”. It is unclear to me what narrative they are referring to, so the authors need to be much more specific.

We’ve clarified this statement in the discussion.

Reviewer #2: The submitted article offers interesting insight into the consequences of SBIs among PWID. A particular strength is the use statewide data. With that, my comments are as follows:

Minor comments:

1. Please ensure you define each acronym when it is first used.

Thank you, we have made changes to define acronyms the first time they are used.

2. With the exception of stratifying by age category, you have quite a large sample. Chi-square is very sensitive to large sample sizes. Because of this, the p-values may be meaningless.

We have removed the P-values from the tables.

3. The cost analysis appears to be very general. While interesting, a deeper look into excess cost attributable to IDU would be far more impactful.

We now present the cost data for SBI among those without drug use diagnosis codes. We’ve included revisions to the methods, results, and discussion detailing these data.

Major comments:

1. In the Statistical Analysis section of the Methods, the explanation for prioritizing SBIs of greatest severity to prevent duplicating cost or counts is not clear. Hospitalizations with multiple SBI would be obscured intentionally then?

We now provide a reference for this practice of prioritizing SBIs and a statement to clarify the methods to prevent count and cost duplication. 

2. For your age category variable, why set the cut off at 60? This age group represents a sizable proportion of your sample (26%). It would be good to know what the breakdown is within this category.

We now provide further breakdown of those 60 or greater, including those who are 60-69, 70-79, and 80 and older.

3. The last sentence of the second paragraph in the Case Classification section ("While SBI hospitalizations.."), are there any published articles to support this?

We now provide a reference for this statement.

4. You indicate that you did not restrict your analysis by age to "capture the full extent of the impact of IDU-related SBI." Further, you offer some deeper justification for including young individuals. However, you do not mention the other end of the age spectrum. We know that older individuals also can be impacted by SBIs unrelated to IDU, and many may have non-illicit, medicinal dependencies. It is possible, if not probable, that a considerable number of older individuals in your sample are misclassified as PWID, yet there is not a single mention of this potential bias. Compounding the issue is that older individuals comprise a large proportion of your sample (26% 60 and older), yet we can't tell if these are mostly people in their 60s, 70s, 80s, etc. All of these issues culminate in a potentially biased cost analysis that likely includes a considerable number of misclassified, older individuals who may incur greater costs in the first place due to infirmity and/or longer lengths of stay.

We know provide further breakdown of those 60 or greater, including those who are 60-69, 70-79, and 80 and older. We now introduce the potential for bias among older patients in the limitations of our manuscript.

---

## [Decision Letter · Decision Letter 1]

28 Oct 2020

Population-based trends in hospitalizations due to injection drug use-related serious bacterial infections, Oregon, 2008 to 2018

PONE-D-20-23516R1

Dear Dr. Menza,

We’re pleased to inform you that your manuscript has been judged scientifically suitable for publication and will be formally accepted for publication once it meets all outstanding technical requirements.

Kind regards,

Nickolas D. Zaller

Academic Editor

PLOS ONE

Additional Editor Comments (optional):

Reviewers' comments:

Reviewer's Responses to Questions

**Comments to the Author**

1. If the authors have adequately addressed your comments raised in a previous round of review and you feel that this manuscript is now acceptable for publication, you may indicate that here to bypass the “Comments to the Author” section, enter your conflict of interest statement in the “Confidential to Editor” section, and submit your "Accept" recommendation.

Reviewer #1: All comments have been addressed

Reviewer #2: All comments have been addressed

2. Is the manuscript technically sound, and do the data support the conclusions?

Reviewer #1: Yes

Reviewer #2: Yes

3. Has the statistical analysis been performed appropriately and rigorously? 

Reviewer #1: Yes

Reviewer #2: Yes

4. Have the authors made all data underlying the findings in their manuscript fully available?

Reviewer #1: Yes

Reviewer #2: No

5. Is the manuscript presented in an intelligible fashion and written in standard English?

Reviewer #1: Yes

Reviewer #2: Yes

6. Review Comments to the Author

Reviewer #1: (No Response)

Reviewer #2: Thank you for addressing the comments from the previous submission. This is an important and interesting article, and I appreciate the opportunity to review it.

7. PLOS authors have the option to publish the peer review history of their article (what does this mean?). If published, this will include your full peer review and any attached files.

Reviewer #1: **Yes: **Robert Heimer

Reviewer #2: No

---

## [Editor Report · Acceptance letter]

29 Oct 2020

PONE-D-20-23516R1 

Population-based trends in hospitalizations due to injection drug use-related serious bacterial infections, Oregon, 2008 to 2018 

Dear Dr. Menza:

I'm pleased to inform you that your manuscript has been deemed suitable for publication in PLOS ONE. Congratulations! Your manuscript is now with our production department. 

Kind regards, 

on behalf of

Dr. Nickolas D. Zaller 

Academic Editor

PLOS ONE